# DOES LLM DREAM OF DIFFERENTIAL EQUATION DISCOVERY?

## ABSTRACT

Large Language Models (LLMs) show promise in symbolic regression tasks. However, applying them to partial differential equation (PDE) discovery presents significant challenges. Unlike traditional symbolic regression, which allows for quick feedback by directly generating data, PDE discovery involves solving implicit equations and deriving data from physical fields, capabilities LLMs currently lack. Our method bridges the gap between LLMs' theoretical understanding of differential equations from textbooks and the practical needs of scientific discovery, where textbooks are less helpful. We show that when physical field data are appropriately formatted and coupled with code generation prompts, general-purpose LLMs can effectively engage in the equation discovery process, even without specific training for this task. This research lays the groundwork for utilizing pre-trained LLMs in automated scientific discovery, while recognizing current limitations and the necessity of hybrid human-AI validation.

## 1 INTRODUCTION

The field of symbolic regression for partial differential equations (PDEs), starting from PDE-FIND Rudy et al. (2017), has experienced remarkable innovation Brunton & Kutz (2024), driven by the convergence of Large Language Models (LLMs) Lorsung & Farimani (2024), advanced evolutionary algorithms Ivanchik & Hvatov (2025); Chen et al. (2022), and physics-informed neural approaches Sun et al. (2025). The most significant trend is the emergence of hybrid methodologies that combine LLM scientific knowledge with evolutionary robustness, fundamentally changing how we approach automated equation discovery from complex data.

In symbolic regression, we observe a paradigm shift from combinatorial search to knowledge-guided generation. While evolutionary algorithms, such as PySR Cranmer (2023), navigate vast hypothesis spaces, LLM-based approaches, like LLM-SR Shojaee et al. (2024), leverage pre-trained scientific knowledge. The main differences in restoring an expression from a differential equation are that the model generates data by itself, i.e., we have direct (and fast) feedback from the model for the evolutionary algorithm or for LLM. For LLM, we note that significant advances have been made by shifting the problem toward the code-generation domain Wang et al. (2025).

In differential equation discovery, we try to find an implicit expression in the form of a differential equation. In contrast, (a) we cannot extract the data field directly from the differential equation (we basically have to "solve" the equation somehow, which is generally a problem by itself) and (b) we cannot easily unpack differential symbols into the data domain. The latter means that if the equation solver is not used, differentiation should be performed within the differential equation discovery algorithm using numerical methods to indirectly assess the discovered equation.

Recent algorithms solve the problem basically in two directions. First, we discuss the numerical difference and how to mitigate the errors associated with it. We use neural networks to filter the data and also differentiate them Du et al. (2024b) as weak forms to make the error weight-averaged Stephany & Earls (2024). For example, classically, the error at the boundaries is larger than that within the bounded domain. The second direction is to develop a differentiation "agnostic" algorithm that can solve equations to form feedback. The solution to the differential equation, in general, makes the process unarguably slower but more robust.

The ambiguity of equation solution necessity creates a fundamental validation crisis - without automated methods to assess physical plausibility, discovered equations require extensive human expertise to verify, thereby limiting their practical deployment. For example, Shojaee et al. (2025) shows the 31.5 % of quality at the top in the physical data for all models. However, recent advances in neural operators show that differentiation (and sometimes the whole differential equation solution process) could be learned Hao et al. (2024). The recent paper demonstrates that a solution can be obtained without relying on solvers Herde et al. (2024). However, most of the success is achieved when we make numerous preliminary assumptions and rely on them.

The LLM, as a portmanteau for any problem nowadays, theoretically contains a wealth of knowledge about differential equations. It can effectively cite and apply the knowledge from the textbook Grayeli et al. (2024). However, differential equation discovery operates on physical field data, creating significant challenges: any LLM or VLM does not often meet the physical data "pictures" (equation solutions) in the training dataset. It is not able to handle the differentiation of such data out of the box. There is initial research on how LLM could be adjusted to the equation discovery problem Du et al. (2024a).

In this paper, we **aim** to test these abilities of general LLM. We formulate differential equation discovery as a code generation problem, and we develop an optimal format of data that enables LLM to extract the connection between differentials and data. The data must not be too compressed to retain physics, and on the other hand, must be compressed to fit the context. Ultimately, we utilize LLM as an oracle to infer the initial possible forms of the equation, which are then passed to the algorithm in a meta-learning loop.

**Contribution**: We formulate PDE discovery as a code-generation task for LLMs, introduce a compact physics-preserving textual representation for field+derivatives, and integrate LLMs as oracles inside an EPDE meta-learning loop.

**Limitation**: - We consider the EPDE single equation discovery framework. However, it could be replaced if necessary. Essentially, we need to find a way to pass the string form into the algorithm, which is a technical task.
- The models with large context, fine-tuned models, etc., may perform better. It is actually a separate task to find a physics-aware pre-training. We use only the publicity available pre-trains.
- We consider only grid-spaced data without missing values. However, we use noise to simulate the real-case scenario.

**Code and data** are available in the GitHub repository `https://anonymous.4open.science/r/EPDE_LLM-2028/`

## 2 DIFFERENTIAL EQUATION DISCOVERY BACKGROUND

In all cases for the equation discovery problem, it is assumed that the data are placed on a discrete grid $X = \{x^{(i)} = (x_1^{(i)}, ...x_{\dim}^{(i)})\}_{i=1}^{i=N}$, where $N$ is the number of observations and dim is the dimensionality of the problem. We mention a particular case of time series, for which $\dim = 1$ and $X = \{t_j\}_{i=1}^{i=N}$.

It is also assumed that for each point on the grid, there is an associated set of observations $U = \{u^{(i)} = (u_1^{(i)}, ..., u_L^{(i)})\}_{i=1}^{N}$ to define a grid map $u : X \subset \mathbb{R}^{\dim} \to U \subset \mathbb{R}^L$.

There are two further ways. First is when we formally determine symbols in form:

$$J^r = (x_1, ..., x_{\dim}; u; D_1u; D_2u; ...; D_ru) \tag{1}$$

,where $D_r = \bigcup_{|\alpha|=r} \{\frac{\partial^r u}{\partial x_1^{\alpha_1}...\partial x_r^{\alpha_r}}\}$ is the set of all partial differentials of order $r$ and $\alpha = \{\alpha_1, ...\alpha_{\dim}\}, |\alpha| = \sum_{i=1}^{i=\dim} \alpha_i$ is just a differential multi-index. *Simply speaking*, equation 1 is a set of symbols that represent differentials up to a given order $r$. Since we usually have a single observation set $u$ we omit it from the notation $J^r(u)$

From these symbols, we get a formal symbolic expression using a possible set of actions $\mathcal{T}$ (monomials, products, powers) acting on $J^r$. Then $S \subset \mathcal{T}$ represents selected terms (equation structure), and $P$ is the set of admissible coefficients. Coefficients by themselves could be a function of independent coordinates or just constants. Then the equation has the following form:

$$M(S, P) = \sum_{s \in S} p_s \cdot s(J^r) = 0 \tag{2}$$

The described process has two differences from symbolic regression: we have an implicit dependency in the form of the equation $M(S, P) = 0$, and also, this equation is differential. To assess any quality measure, we must use a solver to extract a solution from equation 2 and then compare it with the data $U$ on a grid $X$. Using a solver is a computationally intensive approach, even for non-differential expressions; however, for differential equations, it also requires expert solver tuning.

Second way is to use numerical differentiation $D_h$ of data $\bar{J}^r = \{(x^{(i)}, u^{(i)}, D_h u^{(i)}, ...(D_h)^r u^{(i)})\}_{i=1}^{i=N}$. In this case, we can replace symbols with their numerical counterparts, which are essentially tensors of the same dimensionality as the input data. Therefore, numerical differentiation is used to form a resulting tensor that can be used to indirectly assess the equation, for example, by using the mean error, which in the case of the equation is referred to as discrepancy.

For the SINDy case, we manually determine the longest sentence $\Sigma_{\text{long}}$ possible and fix it. The optimization is performed only by $P$, which is essentially a vector of the numerical coefficients near each word of $\Sigma_{\text{long}}$. We need to make $P$ as sparse as possible, which is done with classical LASSO regression. In SINDy, we compute the loss function by using the discrepancy over the discrete grid.

$$P^* = \underset{P \in \Pi}{\text{argmin}} ||M(\Sigma_{\text{long}}, P)||_2 + \alpha ||P||_1 \tag{3}$$

In equation 3 we denote by $|| \cdot ||_2$ the mean discrepancy in the computation grid $X$ and by $|| \cdot ||_1$ is the $l_1$ norm. Since SINDy usually works with constant coefficients, we could use the $l_1$ norm to determine the sparsity of the set of parameters $P$. In some sense, it is a measure of the complexity of the surface in terms of the number of symbols needed to describe it.

Evolutionary approaches and reinforcement learning have their own rules to construct $S$ for a model. Every equation $S_i$ appearing within the optimization process is evaluated using the SINDy approach equation 3 with discrepancy or, as is done in EPDE, by constructing the Pareto frontier over the discrepancy and complexity criteria. Both discrepancy computation and Patero frontier formation are performed as part of the fitness function computation or to generate a reward for the reinforcement learning agent.

There are also more robust measures. For a given surface $M(S, P)$, we try to restore the continuous function $u$ that exactly generates the surface and then compare it with observations $U$. It, of course, requires the solution of the equation. We note that in this case, we do not need to consider jets $J^r$; instead, we begin working with the fibers $u$ and no longer need to consider the differentials $D_r$. In that case, all surfaces are single-connected, i.e., the solution of the equation is unique, which is, of course, a limitation, but it is more robust than a discrepancy measure.

There are also some intermediate cases, such as PIC. Here we spatially handle jets, but temporally restore continuous paths. It could be considered as jet factorization and partial fiber projection.

## 3 DIFFERENTIAL EQUATION DISCOVERY PIPELINES

In this paper, we focus on the differential equation discovery part. That means we do not use a solver to handle the equation, thereby avoiding the need for tuning. Additionally, we do not focus solely on differentiation. All differential fields are obtained equally for both evolutionary algorithms and LLMs. As a result, we pass only the observation data field and differentials to the algorithm to assess its ability to form an equation with indirect equation quality.

We compare the performance of three distinct algorithms (see Fig. 1): the purely evolutionary EPDE framework, the LLM-based discovery approach, and a novel hybrid EPDE+LLM pipeline. The

EPDE framework optimizes equation structures through evolutionary principles, treating each equation as an individual subject to mutation and crossover. In contrast, our LLM method relies on generative symbolic reasoning. The hybrid EPDE+LLM approach is a sequential pipeline: the LLM first generates an initial population of candidate equations, which is then refined by the EPDE algorithm using its evolutionary operations. The following sections delve into the specifics of each method.

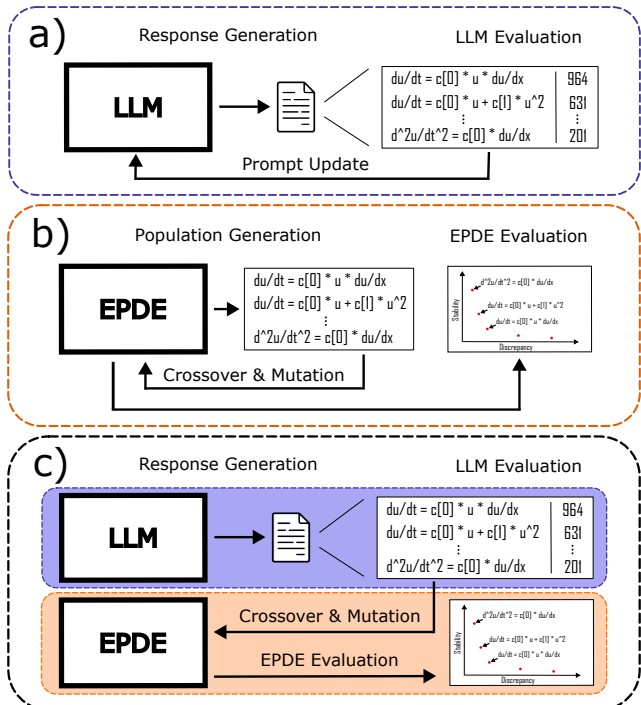

Figure 1: An overview of compared algorithms: a) LLM-based approach, b) EPDE-based approach, c) joint EPDE+LLM approach.

### 3.1 INPUT DATA FIELD PREPARATION

Presenting the raw, high-dimensional data fields directly to the LLM was infeasible due to constraints on the context window. To address this, we evaluated several strategies, including visual language models (VLMs), alternative data transformations, and tensor decomposition techniques. The most effective and viable solution was found to be a significant but careful dimensionality reduction. The original data was downsampled via interpolation to a coarse spatial resolution of approximately 20×20 to 30×30 grid points. This approach preserves the essential structural information of the physical fields while drastically reducing token consumption, making the data tractable for LLM processing. A preliminary analysis of how VLMs handle such physical data was also conducted.

Critical to the success of PDE discovery is the accurate calculation of partial derivatives. For clean data, derivatives were computed using a spectral method based on Chebyshev polynomials. In cases with significant noise, this method was combined with a Butterworth low-pass filter to suppress high-frequency artifacts before differentiation, ensuring numerical stability. The specifics of how the prepared text and numerical data were formatted for the LLM are detailed in Appendix C.

### 3.2 LLM-GENERATED EQUATIONS PIPELINE OVERVIEW

The inspiration for the algorithm came from Shojaee et al. (2025), where they suggest leveraging LLMs' programming skills to compile the desired equation into a Python function. Similar to Shojaee et al. (2025), we also utilize the equation buffer so that the LLM is aware of which attempts improve the approximation.

In every other aspect, the proposed algorithm differs from the LLM-SR approach. All in all, it includes these stages (depicted in Fig. 2):

(1) Response generation;
(2) Equation extraction;
(3) Evaluation of the extracted equation;
(4) Recompilation of the prompt;
(5) Equation buffer pruning.

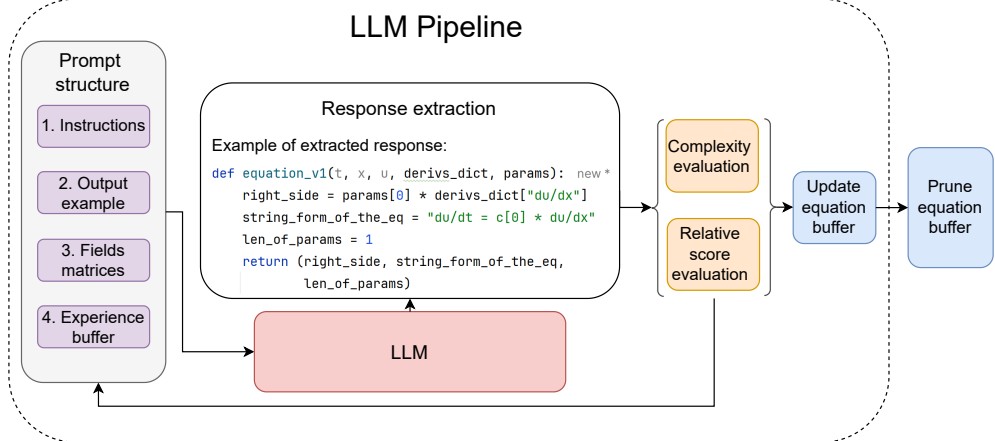

Figure 2: The pipeline of the LLM-based algorithm.

**Response generation** A pivotal factor in this step of the algorithm is prompt engineering. The prompt is divided into the following sections:

1. Instructions. They include problem statement, requirements, and restrictions.
2. A code snippet that defines an evaluator for the LLM-generated solutions.
3. Input data.
4. Experience buffer. Provides the LLM with a performance history of previously proposed equation structures. This buffer, updated iteratively, is implemented as a dictionary where keys are string representations of equations and values are their corresponding relative performance scores (discussed in detail in **Evaluation of the extracted solution** below).
5. An example of input data.

In reality, we use two prompts, depending on the current iteration of the LLM. The prompt for the first iteration is much simpler than those for the subsequent ones, although it also adheres to the structure described above. The second prompt is enhanced, with greater complexity, added constraints, and a refined problem statement.

**Equation extraction** This stage of the algorithm is responsible for extracting, refining, and correcting the solutions generated by the LLM. Despite explicit constraints defined in the prompt, LLM outputs can be unstable and often require post-processing to ensure structural validity and adherence to requirements.

This extraction pipeline significantly improves reliability but cannot guarantee a valid solution in every instance. To ensure overall algorithmic robustness, a failure mode is implemented where iterations containing irresolvable outputs are discarded, following the precedent set by LLM-SR Shojaee et al. (2024).

**Evaluation of the extracted solution** The evaluation mechanism quantifies the quality of an extracted solution through two distinct scores: complexity (Alg. 1) and a relative score (equation 4). The relative score may be defined as a normalized Mean Absolute Error (MAE), assessing predictive accuracy. In contrast, the complexity score evaluates the structural intricacy of the equation based on the number and type of terms that comprise it.

In the proposed algorithm, the left-hand side term $s_{\text{left}}$ is fixed, following the methodology established in the SINDy approach. This design choice was made to initially probe the capabilities of the LLMs under the assumption that the algorithm has correctly identified the balancing term. Each constructed equation is then assigned a normalized Mean Absolute Error (relative score) $R$, defined using the mean $l_2$ norm ($||\cdot||_2$) over all grid points. This score inversely represents quality, with values near 0 indicating high accuracy and a ceiling of 1000 representing the worst-case performance.

$$R = \frac{||M(S, P)||_2}{||s_{\text{left}}||_2} \cdot 1000 \tag{4}$$

The algorithm for complexity evaluation is formalized in App. B. It operates by parsing each equation into its tokens and then assigning a complexity weight based on the token's class and power $p$. The scoring policy is defined as follows: derivative terms are weighted according to $\frac{(n+1)\cdot\beta_d}{2} \cdot p$, where $n$ is the derivative order and $\beta_d$ is a base cost for derivatives. Elementary functions (e.g., sin, cos) incur a cost of $\beta \cdot p$ plus the complexity of their inner terms. Finally, basic variables and constants contribute a cost of $\beta \cdot p$, where $\beta$ is a base cost for simple tokens.

**Recompilation of the prompt**   The prompt provided in App. D is dynamically updated at each iteration to incorporate the latest state of the experience buffer. This buffer serves as a cumulative record of solution performance, implemented as a dictionary where keys are string-based equation descriptors and values are their corresponding relative scores (i.e., normalized mean absolute error, or MAE). The complexity metric is intentionally omitted from this feedback to present the LLM with a single, unambiguous performance objective, as LLMs lack the inherent capability to interpret and optimize within a multi-dimensional fitness space natively.

**Equation buffer pruning**   Following the completion of all iterations, a final refinement stage is applied to the accumulated solution buffer. This stage leverages the previously unused complexity metric to address a key limitation of the relative score: its high sensitivity to noise, which can cause equations with artifacts to outperform correct ones.

To mitigate this, we employ a two-step process. To eliminate the terms that capture noise, we enrich the solution space through a combinatorial expansion. With this method, one of the generated variants is bound to exclude the noisy term, making it highly probable that a correct version of the equation will be discovered.

All equations are then evaluated to form a two-dimensional Pareto front based on complexity and relative score. Finally, a knee detection algorithm identifies the optimal trade-off frontier. The solution space is pruned to retain only those equations lying on or below the calculated supporting line, and the length of the perpendicular distance from this line subsequently ranks these solutions.

### 3.3 EPDE-GENERATED EQUATIONS PIPELINE OVERVIEW

The EPDE (Evolutionary Partial Differential Equation) discovery framework is based on an evolutionary optimization paradigm. A detailed discussion of its capabilities and limitations is available in Maslyaev et al. (2021). Since it is a rather technical detail, we have included it in Appendix A. Main takeaways from the equation discovery algorithm: it can take initial assumptions in the form of an equation string and transform its output back into the string using specific code generation adapters.

### 3.4 JOINT EPDE+LLM PIPELINE OVERVIEW

The joint EPDE+LLM pipeline was designed to leverage the LLM's ability to generate an insightful initial candidate population from the data. This pipeline provides the evolutionary EPDE framework with a high-quality starting population, significantly boosting its capabilities.

As depicted in Fig. 3, the methodology chains together the LLM and EPDE frameworks. The process begins with the LLM generating a broad set of candidate equations. Subsequently, a pruning step enriches this set and then performs filtering to enhance quality. The surviving equations are then mapped into the EPDE framework's representation, serving as the initial population for the final

stage: evolutionary optimization. This stage converges to a Pareto frontier, representing the trade-off between equation accuracy and complexity.

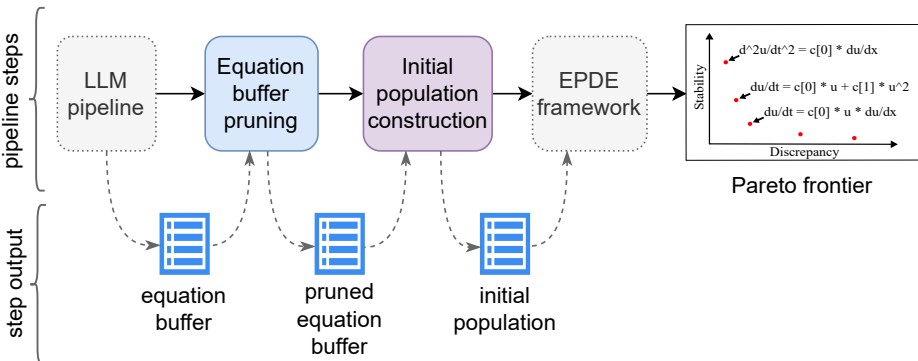

Figure 3: A scheme of joint EPDE+LLM pipeline.

## 4 EXPERIMENTS

The purpose of these experiments is to evaluate the capability of the LLM for equation discovery. Three types of equations — Burgers', wave, and KdV–de Vries — are tested. The datasets used in these experiments are generated numerically. Detailed statements of the initial-boundary value problems and descriptions of the solution methods can be found in App. E. The outcomes are benchmarked against those obtained using the EPDE framework.

All experiments were performed using a `qwen-2.5-72b-instruct` LLM model and the latest EPDE version.

### 4.1 EXPERIMENTAL SETUP

For each experiment, we conduct thirty independent runs of EPDE, LLM-based, and EPDE+LLM frameworks. We evaluate their performance on datasets with and without noise. It is crucial to assess how these frameworks handle noise, as real-world data often contains measurement noise. We use a common approach to add a Gaussian to the data:

$$\tilde{u} = u + \varepsilon \cdot std(u) \cdot N(0,1) \tag{5}$$

where $u$ represents the original clean data, $\tilde{u}$ denotes the noisy data, $N(0,1)$ refers to the standard normal distribution, and $\varepsilon$ indicates the magnitude of the noise.

The magnitudes vary in scale according to the input data. Consequently, each type of equation has a threshold magnitude above which the EPDE fails to identify the target equation in all runs. The noise levels are expressed relative to this threshold and are thus set to the specified noise percentage limit. The maximum permissible noise magnitudes are established as $5 \times 10^{-4}$ for the Wave equation, $2.5 \times 10^{-2}$ for Burgers A, $2 \times 10^{-2}$ for Burgers B, and $5 \times 10^{-3}$ for the Korteweg-de Vries equation. It is worth noting that in experiments involving noisy data, each run is assigned a unique noise profile.

The performance of the algorithm is evaluated using several quality metrics: the discovery rate of the correct equation and the relative error between the coefficients of the identified equations and those of the theoretical model (ground truth). The run becomes successful if at least one strictly correct equation is found.

When multiple solutions are obtained (in the Pareto frontier case), the structure of the equation is verified first. Only if the structure matches correctly is the relative error in coefficients calculated.

We measure the relative error of coefficients using the formula:

$$E(\hat{\xi}_i) = \frac{1}{N} \sum_{i=1}^{N} \frac{|\hat{\xi}_i - \xi_i^*|}{|\xi_i^*|} \tag{6}$$

where $N$ is the total number of terms in the equation, $\hat{\xi}_i$ s the coefficient identified in the discovered equation, and $\xi_i^*$ represents the corresponding coefficient in the true equation.

Furthermore, the hyperparameters used in all experiments are detailed in the supplementary material in App. F. The exact prompts used are listed in App. D.

In the following, we show aggregated tables; more detailed experimental results are in App. G

### 4.2 CLEAN DATA PERFORMANCE COMPARISON

The performance of the EPDE, standalone LLM, and hybrid EPDE+LLM frameworks on clean data is summarized in Table 1. The metrics of interest are the discovery rate (DR), where a higher value is better, and the complexity error (CE), where a lower value is better. The hybrid EPDE+LLM framework consistently achieved the highest discovery rate across all datasets. Notably, for the challenging KdV dataset, the hybrid method's discovery rate (0.37) was more than double that of the standalone EPDE (0.10) and LLM (0.13) approaches. While the standalone LLM showed a higher DR than EPDE on the Burgers A and B datasets, it did so at the cost of a significantly higher CE.

Table 1: Comparison of performance of the frameworks with clean data

| Dataset | EPDE | | LLM | | EPDE+LLM | |
|---|---|---|---|---|---|---|
| | DR | CE | DR | CE | DR | CE |
| Wave | 0.97 | $\mathbf{7.54 \cdot 10^{-4}}$ | 0.97 | $6.57 \cdot 10^{-2}$ | **1.00** | $\mathbf{7.54 \cdot 10^{-4}}$ |
| Burgers A | 0.53 | $\mathbf{8.57 \cdot 10^{-5}}$ | 0.86 | $3.94 \cdot 10^{-4}$ | **0.90** | $\mathbf{8.57 \cdot 10^{-5}}$ |
| Burgers B | 0.50 | $\mathbf{4.55 \cdot 10^{-4}}$ | 0.53 | $9.05 \cdot 10^{-3}$ | **0.90** | $\mathbf{4.55 \cdot 10^{-4}}$ |
| KdV | 0.10 | $\mathbf{1.54 \cdot 10^{-2}}$ | 0.13 | $1.92 \cdot 10^{-2}$ | **0.37** | $\mathbf{1.54 \cdot 10^{-2}}$ |

The results demonstrate that integrating LLM-generated candidate equations into the EPDE search process robustly enhances discovery performance. The LLM framework serves as an effective hypothesis generator for equation structures, while the EPDE methodology provides refined numerical optimization for parameter identification.

### 4.3 NOISY DATA PERFORMANCE COMPARISON

The performance of the frameworks under significant noise levels (25% to 100%) is presented in Table 2. The hybrid EPDE+LLM framework demonstrates superior robustness, achieving the highest discovery rate in 10 out of 16 dataset-noise combinations. A key observation is that the LLM's contribution is not contingent on its ability to find the correct equation itself. For instance, on the Wave and Korteweg-de Vries equations, the standalone LLM failed (DR = 0.00 across most noise levels). Nevertheless, its equation suggestions substantially improved the performance of the hybrid EPDE+LLM model, indicating that the LLM acts as an effective generator of meaningful candidate equations, even when its own symbolic regression fails to do so.

An interesting anomaly is observed for the Burgers A dataset, where the LLM-based approach outperforms both EPDE and the hybrid approach. For this specific equation, the LLM's search strategy is less susceptible to a local minimum that traps the EPDE algorithms—a phenomenon where an incorrect equation form achieves a deceptively optimal objective function value given the noisy data. Despite this, the hybrid approach maintains competitive performance across the other three datasets, confirming its overall robustness.

While the discovery rate indicates the frequency of finding the correct equation form, the accuracy of the identified coefficients is equally critical. Table 3 presents the mean coefficient errors (in units of $10^{-4}$) alongside their standard deviations, providing a complementary view of performance. The results reveal that a high discovery rate does not always guarantee precise parameter estimation. For instance, on the Burgers B dataset at 50% noise, the standalone LLM achieves a high DR of

Table 2: Comparison of discovery rates of the frameworks with noisy data

| Noise level | Framework | Dataset | | | |
|---|---|---|---|---|---|
| | | Wave | Burgers A | Burgers B | KdV |
| 25% | EPDE | 0.17 | 0.20 | 0.17 | 0.10 |
| | LLM | 0.00 | **0.73** | 0.63 | 0.06 |
| | EPDE+LLM | **0.73** | 0.26 | **0.66** | **0.57** |
| 50% | EPDE | 0.17 | 0.23 | 0.10 | 0.23 |
| | LLM | 0.00 | **0.73** | **0.50** | 0.00 |
| | EPDE+LLM | **0.36** | 0.10 | 0.30 | **0.40** |
| 75% | EPDE | 0.07 | 0.13 | 0.07 | 0.13 |
| | LLM | 0.00 | **0.76** | 0.07 | 0.00 |
| | EPDE+LLM | **0.23** | 0.16 | **0.16** | **0.30** |
| 100% | EPDE | 0.03 | 0.03 | 0.03 | 0.03 |
| | LLM | 0.07 | **0.80** | 0.07 | 0.00 |
| | EPDE+LLM | **0.20** | 0.23 | **0.20** | **0.30** |

0.50 (Table 2), but its coefficient error is significantly larger than that of the EPDE+LLM hybrid, indicating less stable and accurate parameter fits. Conversely, the hybrid EPDE+LLM framework demonstrates remarkable consistency; its leading or competitive discovery rates are often paired with the lowest or most stable coefficient errors, as seen prominently in the Wave and KdV datasets. The LLM+EPDE hybrid has dual advantage: it not only finds the correct equation structure more reliably but also converges to more accurate and robust parameter estimates, a crucial characteristic for practical applications with noisy data.

Table 3: Comparison of coefficient errors $(10^{-4})$ of the frameworks with noisy data

| Noise level | Framework | Dataset | | | |
|---|---|---|---|---|---|
| | | Wave | Burgers A | Burgers B | KdV |
| 25% | EPDE | 40.7±5.00 | 44.0±11.0 | 39.0±2.96 | 1334±4775 |
| | LLM | - | **17.4±1.39** | 56.9±1.42 | 1778±106 |
| | EPDE+LLM | **40.4±3.51** | 37.7±28.5 | **27.3±2.66** | **169±65.0** |
| 50% | EPDE | 8.42±2.31 | 162±13.7 | 242±22.2 | **298±1.39** |
| | LLM | - | **4.17±1.55** | 400±3.77 | - |
| | EPDE+LLM | **5.92±2.65** | 95.7±12.3 | **242±9.33** | 326±0.82 |
| 75% | EPDE | **8.33±65.4** | 358±51.8 | 532±63.8 | **282±2.71** |
| | LLM | - | **37.9±5.17** | 4997±3.94 | - |
| | EPDE+LLM | 13.9±2.50 | 212±4.51 | **520±3.50** | 289±51.4 |
| 100% | EPDE | 998 | 576 | **858** | **262** |
| | LLM | 2546±809 | **86.1±6.20** | 4967±8.69 | - |
| | EPDE+LLM | **18.2±11.2** | 376±12.3 | 1206±953 | 291±2.71 |

These complementary strengths suggest promising avenues for integrating the framework. A hybrid methodology that leverages LLMs' structural discovery capabilities for initial equation identification, followed by EPDE's precision optimization for parameter refinement, could yield superior overall performance in noisy environments. This synergistic approach would combine the noise resilience of linguistic processing with the precision of evolutionary computation, potentially addressing the limitations observed in both individual frameworks.

## 5 CONCLUSION

The trivial results are that LLM could be used to replace evolutionary optimization. It has its own advantages and drawbacks. With proper instruction, for example, it can generate compact forms, as is partially done in PDE-READ. However, apart from the success of structural optimization, there is a failure in determining the numerical coefficient.

We show that EPDE+LLM form a practical, complementary pair: we pass a small field snapshot to an LLM to generate compact structural hypotheses, then pass the full dataset and a simple initial coefficient guess to EPDE for numerical differentiation, structure refinement, and coefficient fitting. This two-stage workflow narrows the search space and yields cleaner, more reliable discovered PDEs than either component alone. We did not evaluate the LLM for numerical differentiation and do not expect it to replace dedicated numerical modules, which remain necessary for accurate residual evaluation and coefficient estimation.

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

## A EPDE ALGORITHM DETAILS

This section provides a brief overview of the algorithm's core evolutionary operators: mutation and crossover, as well as the internal equation representation and fitness evaluation scheme.

**Model definition** Evolutionary algorithms construct model structures through the application of elementary operations. To minimize the computational cost associated with structural optimization, the EPDE framework utilizes building blocks known as *tokens*. These tokens represent parametrized families of functions and operators. A token is formally defined by the equation 7.

$$t = t(\pi_1, ...\pi_n) \tag{7}$$

In equation 7, the symbols $\pi_1, \ldots, \pi_n$ denote the parameters of the token, which will be elaborated below.

To differentiate between an individual token and a product of tokens—referred to as a *term* — we introduce the notation $T = t_1 \cdot ... \cdot t_{T_{length}}$, where the term length $T_{length}$ satisfies $0 < T_{length} \leq T_{max}$. Here, $T_{max}$ is a hyperparameter of the algorithm. It is crucial to note that although $T_{max}$ influences the final form of the discovered model, a reasonable value for the number of tokens per term (typically 2 or 3) is often sufficient to represent the structure of most actual differential equations.

Tokens $t_i$ are organized into *token families* $\Phi_j$ to facilitate finer control over the model's form. All tokens within a given family share a fixed set of parameters $\pi_1, \ldots, \pi_n$. For instance, a family of differential operators can be defined as $\Phi_{der} = \{\frac{\partial^{\pi_{n+1}} u}{\partial^{\pi_1} x_1 ... \partial^{\pi_n} x_n}\}$ to enable the discovery of linear or nonlinear equations with constant coefficients. Similarly, a trigonometric token family $\Phi_{trig} = \{\sin(\pi_1 x_1 + ... + \pi_n x_n),$
$\cos(\pi_1 x_1 + ... + \pi_n x_n)\}$ can be introduced to search for forcing functions or variable coefficients.

Token parameters can be either optimizable or non-optimizable. It is often advantageous to fix certain parameters, grouping tokens with identical fixed values into a single family as non-optimizable entities. This approach allows, for example, differential tokens to appear multiple times within a single term to represent nonlinearity. In contrast, trigonometric tokens are typically optimized and, if required, appear only once per term. The algorithm accepts as input the unified set $\Phi = \bigcup_j \Phi_j$ of token families, which is specified by the user.

For simplicity, we operate under the assumption that all tokens are pre-computed on a fixed discrete grid. The specific choice of grid does not affect the fundamental description of the algorithm. Consequently, the structure of the equation and the parameters of its tokens remain the sole variables in the differential equation model presented in equation 8.

$$M(S, \{C, P\}) = \sum_{j=1}^{j \leq N_{terms}} C_j T_j \tag{8}$$

In equation 8, the structure $S$ comprises a set of terms $\{T_j\}_{j=1}^{j=N_{terms}}$, each constructed from a product of distinct tokens. The model parameters are partitioned into two sets: (1) the term coefficients $C = \{C_j\}_{j=1}^{j=N_{terms}}$, where each $C_j$ is a scalar coefficient for term $T_j$, and (2) the optimizable parameters $P = \{\pi_1, ...\}$ of variable length. The composition and cardinality of $P$ may differ for each model and can be modified by the evolutionary operators during the optimization process.

The maximum number of terms, $N_{terms}$, is a hyperparameter of the algorithm. It is important to note that $N_{terms}$ serves not a directive but a restrictive function. The actual number of terms in the final model may be less than $N_{terms}$, as it is subject to reduction through the fitness-based selection procedure described below.

To facilitate visualization of the following evolutionary operator schemes (Fig. 4), we employ a simplified individual representation. Each individual in this context corresponds to an instance of the model defined in equation 8.

$$C_1 \cdot \boxed{u^2 \frac{\partial^2 u}{\partial x \partial t}} + C_2 \cdot u \cdot \boxed{\frac{\partial u}{\partial t}} \cdot \boxed{\frac{\partial u}{\partial x}} + C_3 \cdot \boxed{\frac{\partial^2 u}{\partial t^2}} + C_4 \cdot \boxed{\frac{\partial^3 u}{\partial x^3}} \cdot u = 0$$

$T_1 \qquad t_1 \quad t_2 \quad t_3 \qquad T_3 \qquad T_4$

$N_{terms}$

| $T_1$ | $T_2$ | $T_3$ | $T_4$ |
|---|---|---|---|

| $t_1$ | $t_2$ | $t_3$ |
|---|---|---|

$T_{length}$

Figure 4: Model visualization: $T_i$ are the token products from equation 8 and $t_i$ are the tokens from equation 7.

The optimization process is conducted in two stages: structural and parametric. The population is initialized following equation 8 and possessing distinct, randomly generated structures. After the initialization step, the parametric optimization stage computes a fitness value for each individual.

**Fitness evaluation** Fitness evaluation fulfills two objectives: (1) determining the parameters $\{C, P\}$ for each model, and (2) providing a standard fitness metric. The evaluation procedure involves selecting one term from the structure $S$ as a "target", which requires transforming the individual model into the form given by equation 9 prior to fitness computation.

$$T_{target} = \sum_{j=1,...target-1}^{j=target+1,...,N_{terms}} C_j T_j \tag{9}$$

The variable $target$ in equation 9 represents a randomly selected index. This random selection prevents the algorithm from converging to the trivial solution where $\forall j \ C_j = 0$. For the purpose of fitness computation, the terms $T_j$ are held fixed. The objective is to determine the coefficients $C = C_1, ...C_{target}, ...C_{N_{terms}}\}$ and the optimizable parameters $P = \{P_1, ...P_{target},$ $...P_{N_{terms}}\}$ (if they exist). A key constraint is that $C_{target} \equiv -1$, and the parameters in the set $P_{target}$ are always fixed.

The optimal term coefficients $C_{opt}$ and the optimal parameter sets $P_{opt}$ are computed using LASSO regression, as formalized in equation 10.

$$C_{opt}, P_{opt} = \arg \min_{C,P} \left\| T_{target} - \sum_{j=1,...target-1}^{j=target+1,...,N_{terms}} C_j T_j \right\|_2 +$$
$$+ \lambda (\|C\|_1 + \|P\|_1)\} \tag{10}$$

In equation 10, $\|\cdot\|_p$ designates the $l_p$ norm. After performing LASSO regression, coefficients are compared to a minimal coefficient value threshold. Terms with $|C_j|$ below this threshold are removed, thereby refining the model and preventing the excessive growth of redundant terms.

After obtaining the final set of optimal coefficients from equation 10, the fitness function $F$ is calculated as defined in equation 11.

$$F = \frac{1}{\left\lVert M(S, \{P_{opt}, C_{opt}\})\big|_X \right\rVert_2} \tag{11}$$

In essence, the denominator in equation 11 represents the average discrepancy over the computation grid $X$.

**Evolutionary operators**  To ensure valid equation generation in the population initialization step, cross-over and mutation operators, expert rules are designed for each $S_{ind}$. These rules prevent ill-formed equations (e.g., $0 = 0$) and redundant terms, such as those generated by commutative multiplication, without constraining the overall solution space. Each model structure $S_{ind}$ has an associated set of forbidden tokens that are excluded from crossover and mutation events.

The selection of tokens during mutation and the exchange of terms during crossover are both equiprobable, with the only limitation presented by expert rules.

The crossover operator is defined as the exchange of terms between two individuals, as illustrated in Fig. 5. The terms selected for this exchange are chosen from a uniform distribution, meaning every term has an equal probability of being involved.

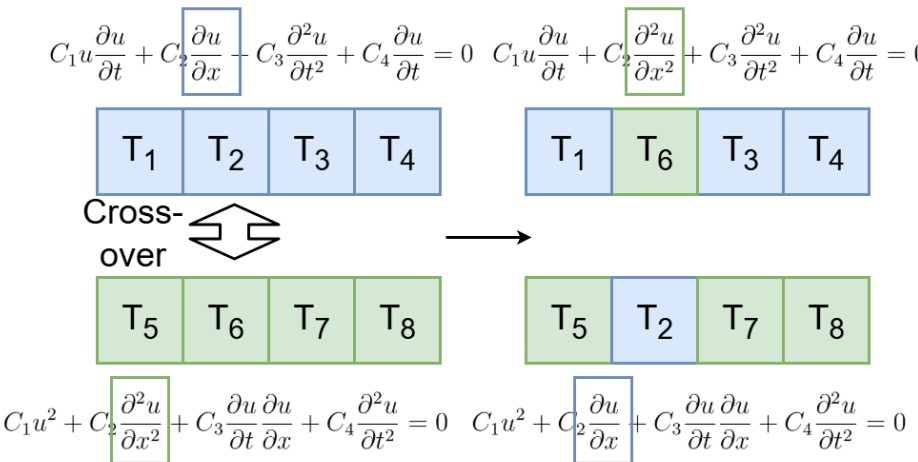

Figure 5: EPDE cross-over operator scheme.

The mutation operator, demonstrated in Fig. 6, has two modes: term exchange and token exchange.

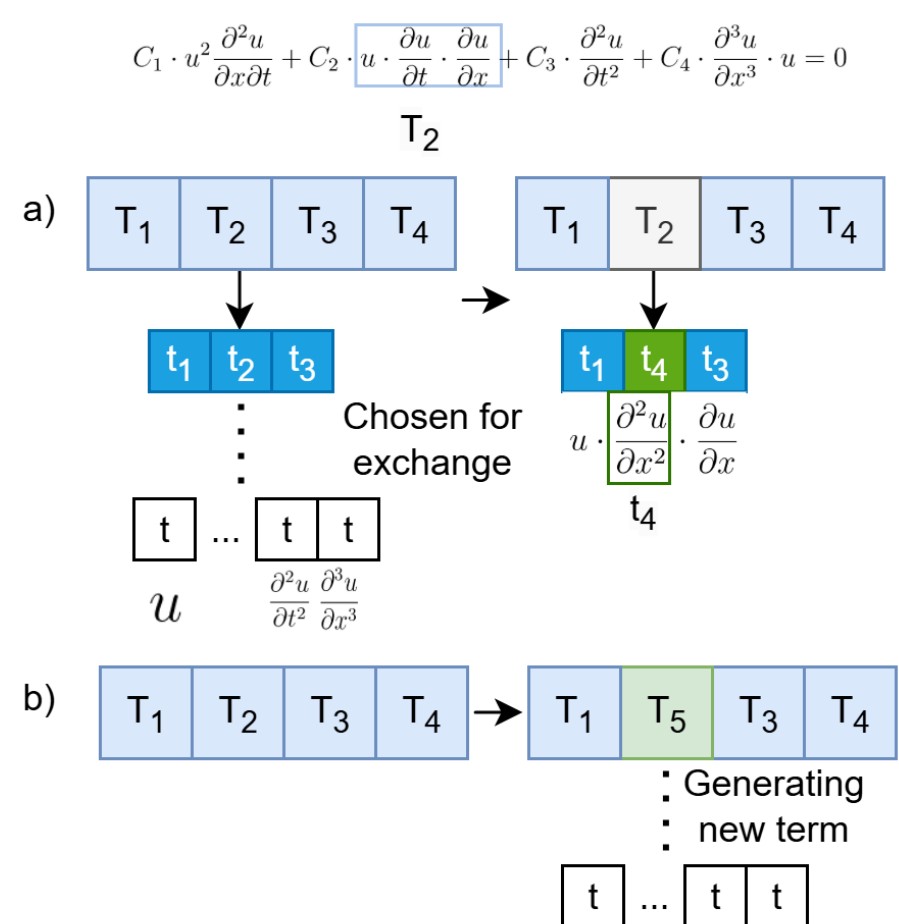

Figure 6: EPDE mutation operator scheme.

As shown in Fig. 6a), token exchange replaces one token with another from a homogeneous pool. Term exchange (Fig. 6b) generates a new term from the same pool by first randomly selecting a term length and then populating it with tokens chosen uniformly from the pool.

To summarize, the inputs are the observational data $U$ on grid $X$ and the token families $\Phi$. The output is a differential equation of the form $Lu = f$, whose type (ODE, PDE, or system) depends on the dimensionality of $U$ and $X$.

# B   COMPLEXITY EVALUATION

**Data:** List of terms - $terms$
**Result:** Complexity score
$\beta_d = 0.5$;
$\beta = 0.2$;
$complexity = 0$;
**for** $term$ $in$ $terms$ **do**
    **for** $token$ $in$ $term$ **do**
        $p = extract\_power(token)$;
        **if** $token$ $is$ $derivative$ **then**
            $n = extract\_derivative\_order(token)$;
            $complexity = complexity + \frac{(n+1)\cdot\beta_d}{2} \cdot p$;
        **else**
            **if** $token$ $is$ $function$ **then**
                $complexity = complexity + \beta \cdot p + eval\_complexity(inner\_terms)$;
            **else**
                $complexity = complexity + \beta \cdot p$;
            **end**
        **end**
    **end**
**end**

**Algorithm 1:** The pseudo-code of complexity evaluation

## C INITIAL TESTS ON LLM'S UNDERSTANDING OF THE DATA

The fundamental question of our research was whether Large Language Models (LLMs) could discern functional dependencies within numerical data fields, presented, in our case, as two-dimensional data, and to identify which class of LLMs is best suited for this task. Given the spatial nature of the data, where $u$ is a matrix defined over discrete $x$ and $t$, our initial hypothesis inclined towards visual LLMs (VLLMs), which are designed to process image data.

A series of preliminary experiments, however, demonstrated that these visual LLMs struggled significantly with the core requirement of the task. They exhibited a notable inability to accurately interpret the content of even basic visual representations of the data (see the subsection below). The models failed to reliably identify data values from the heatmaps, let alone discover the underlying mathematical relationships between variables.

In contrast, experiments with textual representations of the data revealed that even small-scale textual LLMs could often propose equation structures that approximated the underlying function. This critical result - that textual models showed a surprising aptitude for the provided task - justified our pivot to textual LLMs and encouraged the development of the current pipeline.

A detailed analysis of these experiments is provided in the following subsections.

### C.1 SPACE PERCEPTION TESTS ON VISUAL LLMS

The tests were performed mainly on the heatmaps derived from functions $cos(C \cdot x)$, chosen for their clear periodic structure, with the exception of the last test which was based on a hypothesis that the problem lies in the nature of the images and not in characteristics of VLLM. The models evaluated were: `gemini-pro-vision`, `qwen-2-vl-72b-instruct`, `llama-3.2-90b-vision-instruct`.

The experimental design, illustrated in Fig. 7, systematically examined different potential failure modes:

- Test (a) and (b) assessed basic pattern recognition ability by varying the frequency of oscillation ($\cos(2.5x)$ and $\cos(10x)$).
- Test (c) hypothesized that the monochromatic color scheme of standard heatmaps might be a limiting factor and tested the same high-frequency function ($\cos(10x)$) with a color mapping.
- Test (d) served as a core control. This test was used as a primal indicator of models' ability to understand periodic structures while accounting for their training data distribution, which consists largely of human-recognizable scenes.

The image resolution was mostly set to $128 \times 128$ pixels. An exception was the control image in case (d), which was rendered at a higher resolution of $512 \times 512$ to ensure clarity. Furthermore, to systematically rule out resolution-based limitations, case (c) was tested across multiple scales: $128 \times 128$, $256 \times 256$, $512 \times 512$, and $1024 \times 1024$. This range of resolutions was selected to test the models' limits, with the baseline set to a low resolution of $128 \times 128$ to reflect the typical scale of our numerical datasets, which does not exceed $512 \times 512$ pixels.

The experimental results revealed significant limitations in the visual LLMs' capabilities. In case (a), they misclassified a cosine gradient as linear and could not correctly count two minima. In the higher-frequency case (b), all models underestimated the count of extrema (reporting 5 or less vs. a true count of 6-7 for each type of extrema). Altering resolution and adding a color mapping in case (c) produced no substantial improvement, with a faint positive effect only at the maximum tested size of $1024 \times 1024$ pixels, where the 7th maximum was sometimes noted. Lastly, in (d) case the models reported the existence of 20 to 30 elements on the image with the only exception of qwen-2-vl-72b-instruct, which correctly identified 25.

These experiments led us to conclude that visual LLMs are ill-suited for this specific task. Consequently, we pivoted to textual LLMs. While raw numerical data is also non-ideal for these models, we hypothesized that a transformation of the data into a suitable textual format could leverage their strengths in symbolic reasoning and pattern recognition.

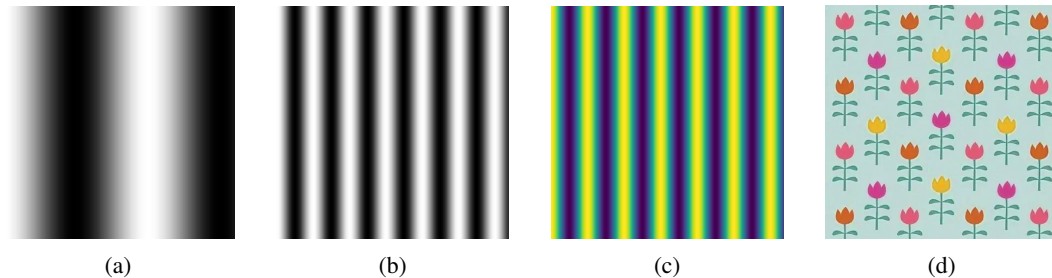

(a)          (b)          (c)          (d)

Figure 7: The input data for space perception tests on Visual LLMs: (a) A heatmap of $cos(2.5x)$, (b) A heatmap of $cos(10x)$, (c) A heatmap of $cos(10x)$ in colours, (d) An image containing an unambiguous periodic pattern of floral elements. The recognizability of these elements to a human observer establishes a baseline for expected model performance.

## C.2 SPACE PERCEPTION TESTS ON TEXTUAL LLMS

To evaluate the inherent pattern recognition capabilities of textual LLMs, we conducted initial experiments on one-dimensional data generated from the function $u(x) = \sin(2.5x)$ (see the prompt template in Appendix D.3). The models tested were `qwen2.5-72b-instruct` and `mixtral-8x7b-instruct`. Both models correctly identified the sinusoidal nature of the function, demonstrating their ability to recognize periodic patterns from numerical data. Although they made errors in estimating the precise oscillation parameters, their successful relation identification provides initial validation of our core hypothesis: that textual LLMs can serve as effective tools for extracting functional relationships from structured numerical data.

We subsequently extended our investigation to two-dimensional functions. The test case was designed to be partially periodic: $u(t,x) = 2\sin(2.5x) + 0.07t^2$. The models tested were `qwen2.5-72b-instruct`, `mixtral-8x7b-instruct`, and the larger `mixtral-8x22b-instruct`. The results were promising yet incomplete. The `qwen2.5-72b-instruct` model, for instance, correctly identified the sinusoidal component along the $x$-dimension in 9 out of 10 trials. While it never explicitly identified the quadratic term $t^2$, it consistently recognized the non-periodic, increasing trend along the $t$-dimension. This demonstrates a capacity for discerning composite spatial structures, albeit with limited parametric precision.

The other models yielded similar results, though `mixtral-8x7b-instruct` performed noticeably worse - as a rule, the model insisted on a polynomial structure, occasionally suggesting a sinusoidal function along $x$ dimention; while the `mixtral-8x22b-instruct` performed on par with `qwen2.5-72b-instruct`, producing responses of equivalent quality and insight.

An essential aspect of our testing involved determining the optimal data representation for textual LLMs. We evaluated two distinct formats:

(1) Structured Tabular Data: A three-column format with headers "x, t, u", where each subsequent row represented a single data point (the prompt is given in Appendix D.4).
(2) Raw array data: The direct string representation of the two-dimensional NumPy array for $u$, provided in a row-major format (the prompt is showcased in Appendix D.5).

This comparison was crucial for assessing whether the models benefited from explicit feature structuring or could infer relationships from raw numerical arrays. The results demonstrated a significant advantage for the structured tabular format. When presented with the "x, t, u" table, the top-performing models (`qwen2.5-72b-instruct` and `mixtral-8x22b-instruct`) successfully identified the sinusoidal relation along the $x$-dimension in approximately 90% of cases (9/10 trials). In contrast, the same models achieved only a 60% success rate (6/10 trials) when the data was presented as a raw numerical array. This clear performance gap underscores the importance of feature-label structuring for enabling textual LLMs to perform spatial reasoning tasks.

# D  PROMPTS

## D.1  PROMPT FOR THE ZEROTH ITERATION OF THE LLM PIPELINE

What is a possible function with the general equation form {full_form} that could be described ↩
    ↪ with the set of points named points_set, that have the form of '{dots_order}'. Give an ↩
    ↪ answer in the function equation_v1 constructing it in a way described by equation_v1 ↩
    ↪ in the end.
Note that although the general equation form is {full_form}, the resulting equation may take on ↩
    ↪ simpler forms, for ex., {left_deriv} = F(t, du/dx) or {left_deriv} = F(du/dx). Suggest ↩
    ↪ some simple structure, that roughly describe the relationships in data, for example ↩
    ↪ {left_deriv} = c[0] * du/dx.

Requirements:
1. Only output your reasoning and the code starting from "def equation_v1..." DO NOT recite ↩
    ↪ the other functions (like loss_function evaluate etc.)

```
import numpy as np
from scipy.optimize import minimize

def loss_function(params, t, x, u, derivs_dict):
    u_pred = equation_v1(t, x, u, derivs_dict, params)[0]
    return np.mean((u_pred−derivs_dict["{left_deriv}"])**2)

def evaluate(data: dict) −> float:
    """ Evaluate the constructed equation"""
    inputs, derivs_dict = data['inputs'], data["derivs_dict"]

    # Optimize equation skeleton parameters
    loss_partial = lambda params: loss_function(params, *inputs, derivs_dict)
    params_initial_guess = np.array([1.0]*P)
    result = minimize(loss_partial, params_initial_guess, method='BFGS')
    optimized_params = result.x

    # Return evaluation score
    score = loss_function(optimized_params, *inputs, derivs_dict)
    return score if not np.isnan(score) and not np.isinf(score) else None

#/Input data

points_set =
{points_set}

#/end of input data

# An example of desired output:
'''python
def equation_v1(t: np.ndarray, x: np.ndarray, u: np.ndarray, derivs_dict: dict(), params: np.ndarray):
    right_side = params[0] * derivs_dict["du/dx"]
    string_form_of_the_equation = "{left_deriv} = c[0] * du/dx"
    len_of_params = 1
    return right_side, string_form_of_the_equation, len_of_params

'''
```

## D.2  PROMPT FOR THE SUBSEQUENT ITERATIONS OF THE LLM PIPELINE

What is a possible function with the general equation form {full_form} that could be described ↩
    ↪ with the set of points named points_set, that have the form of '{dots_order}'? Give an ↩
    ↪ answer in the function equation_v1 constructing it in a way described by the example ↩
    ↪ in the end.
Your goal is to explore the equations space (in relation to their scores) and to examine any ↩
    ↪ inexplicit interactions between the input variables (for ex. du/dx ∗ u^2).
The dictionary exp_buffer stores previous attempts to find the equation evaluated with evaluate ↩
    ↪ function. Refer to it in order to understand what is yet to be explored and what might ↩
    ↪ be worth more exploration. The best score is 0.
Also, keep in mind, if it seems like t or x are involved in the equation do not forget that u and ↩
    ↪ its derivatives are dependent on them, and thus the involvement of t and x might be ↩
    ↪ expressed through u or its derivatives. Your goal is to find any possible inexplicit ↩
    ↪ interactions.
Start by exploring simpler structures and then gradually move on to more complicated ones IF ↩
    ↪ you see the need to do so.

Note that although the general equation form is {full_form}, the resulting equation may take on ↩
    ↪ simpler forms (BUT IT DOESN'T HAVE TO!), like {left_deriv} = F(t, du/dx).
Make sure the suggested equation is dependent on at least one derivative, (e.g, in case of du/dt ↩
    ↪ = F(t, x, u, du/dx), du/dx must be included).

Requirements:
1. First look at the exp_buffer and then suggest the equation, the string form of which is not ↩
    ↪ already there!
2. Do not copy the equations from the exp_buffer!
3. Only give a simplified version of the equation in string_form_of_the_equation: always open ↩
    ↪ the brackets, for ex. instead of 'du/dt = c[0] ∗ (1 + du/dx) ∗ t' return 'du/dt = c[0] ∗ t + ↩
    ↪ c[1] ∗ du/dx ∗ t'.
4. Higher order derivatives must be referenced as d^nu/dx^n or d^nu/dt^n, where n is an integer ↩
    ↪ (for example, d^2u/dx^2 and NOT du^2/dx^2). Anything like du^n/dx^n refer to the ↩
    ↪ multiplication of du/dx and should be written as (du/dx)^n or (du/dx)∗∗n (same apply ↩
    ↪ to du/dt).
5. Do not put {left_deriv} into the right side of the equation as a standalone term, you can ↩
    ↪ though use it as part of a term: ..+ {left_deriv} ∗ u +.. for example

```
import numpy as np
from scipy.optimize import minimize

def loss_function(params, t, x, u, derivs_dict):
    u_pred = equation_v1(t, x, u, derivs_dict, params)[0]
    return np.mean((u_pred−derivs_dict["{left_deriv}"])∗∗2)

def eval_metric(params, t, x, u, derivs_dict, left_side):
    u_pred = equation_v1(t, x, u, derivs_dict, params)[0]
    return np.mean(np.fabs(u_pred − derivs_dict[left_side]))

def evaluate(data: dict) −> float:
    """ Evaluate the constructed equation"""
    inputs, derivs_dict = data['inputs'], data["derivs_dict"]
    # Optimize equation skeleton parameters
    loss_partial = lambda params: loss_function(params, ∗inputs, derivs_dict)
    params_initial_guess = np.array([1.0]∗P)
    result = minimize(loss_partial, params_initial_guess, method='BFGS')
    optimized_params = result.x
    # Return evaluation score
    score = eval_metric(optimized_params, ∗inputs, derivs_dict, left_side)
    return score if not np.isnan(score) and not np.isinf(score) else None
```

#/Input data

points_set =
{points_set}
exp_buffer = {{
}}

#/end of input data

# An example of desired output:
```python
def equation_v1(t: np.ndarray, x: np.ndarray, u: np.ndarray, derivs_dict: dict(), params: np.ndarray):
    right_side = params[0] * derivs_dict["du/dx"]
    string_form_of_the_equation = "{left_deriv} = c[0] * du/dx"
    len_of_params = 1
    return right_side, string_form_of_the_equation, len_of_params

```

### D.3  1D CASE OF TEXTUAL LLMs' TESTING

What is a possible function (e.g. u(x) = x**2 + 5) that could be described
with this set of points, that have the form of "x u(x)":

0.00 0.00
0.21 0.50
0.42 0.87
0.63 1.00
0.84 0.86
1.05 0.49
1.26 −0.02
1.47 −0.52
1.68 −0.88
1.89 −1.00
2.11 −0.85
2.32 −0.47
2.53 0.03
2.74 0.53
2.95 0.88
3.16 1.00
3.37 0.84
3.58 0.46
3.79 −0.05
4.00 −0.54

### D.4  2D CASE OF TEXTUAL LLMs' TESTING WITH STRUCTURED TABULAR DATA

What is a possible function (e.g. u(x, t) = x**2 + 5t) that could be described with this set of ↩
    ↪ points, that have the form of "t x u(t, x)":

0.00 0.00 0.00
0.00 0.21 1.00
0.00 0.42 1.74
0.00 0.63 2.00
0.00 0.84 1.72
0.00 1.05 0.98
0.00 1.26 −0.03
0.00 1.47 −1.03

0.00 1.68 −1.75
0.00 1.89 −2.00
0.00 2.11 −1.70
0.00 2.32 −0.95
0.00 2.53 0.07
0.00 2.74 1.06
0.00 2.95 1.77
0.00 3.16 2.00
0.00 3.37 1.69
0.00 3.58 0.92
0.00 3.79 −0.10
...

### D.5   2D CASE OF TEXTUAL LLMS' TESTING WITH RAW ARRAY DATA

What is a possible function (e.g. u(t, x) = x∗∗2 + 5t) that could be described with this array, that ↩
↪   represents the function u(t, x)":

[[ 0. 1. 1.74 2. 1.72 0.98 −0.03 −1.03 −1.75 −2. −1.7 −0.95
    0.07 1.06 1.77 2. 1.69 0.92 −0.1 −1.09]
  [ 0.02 1.02 1.76 2.02 1.74 1. −0.01 −1.01 −1.73 −1.98 −1.68 −0.93
    0.08 1.08 1.79 2.02 1.71 0.94 −0.08 −1.07]
...

## E   EQUATION PROBLEM STATEMENTS

### E.1   BURGERS A

The initial-boundary value problem for Burger's equation is represented with equation 12.

$$
\frac{\partial u}{\partial t} + u\frac{\partial u}{\partial x} = 0
$$
$$
u(0,t) = \begin{cases} 1000, t \geq 2 \\ 0, t < 2 \end{cases}
$$
$$
u(x,0) = \begin{cases} 1000, x \leq -2000 \\ -x/2, -2000 < x < 0 \\ 0, \text{otherwise} \end{cases} \tag{12}
$$
$$
(x,t) \in [-4000, 4000] \times [0, 4]
$$

The analytical solution to the problem presented in equation 12 is given in Rudy et al. (2017). Data for the experiment were obtained with the discretization of the solution in the domain $(x,t) \in [-4000, 4000] \times [0, 4]$ using $101 \times 101$ points.

### E.2   BURGERS B

The problem and data were provided by the authors of PySINDY[1]. The problem can be formulated in equation 13, where the boundary conditions were not reported. The solution was provided for the domain $(x,t) \in [-8, 8] \times [0, 10]$ using $256 \times 101$ discretization points.

$$
\frac{\partial u}{\partial t} + u\frac{\partial u}{\partial x} - 0.1\frac{\partial^2 u}{\partial x^2} = 0
$$
$$
(x,t) \in [-8, 8] \times [0, 10] \tag{13}
$$

### E.3   KORTEWEG-DE VRIES

As in the case of Burgers' equation, the data and the problem (equation 14) were provided by the authors of PySINDY for the domain $(x,t) \in [-30, 30] \times [0, 20]$ using $512 \times 201$ discretization points.

$$
\frac{\partial u}{\partial t} + 6u\frac{\partial u}{\partial x} + \frac{\partial^3 u}{\partial x^3} = 0
$$
$$
(x,t) \in [-30, 30] \times [0, 20] \tag{14}
$$

### E.4   WAVE

The initial-boundary value problem for the wave equation is given in equation 15.

$$
\frac{\partial^2 u}{\partial t^2} - \frac{1}{25}\frac{\partial^2 u}{\partial x^2} = 0
$$
$$
u(0,t) = u(1,t) = 0
$$
$$
u(x,0) = 10^4 \sin^2 \frac{1}{10}x(x-1)
$$
$$
u'(x,0) = 10^3 \sin^2 \frac{1}{10}x(x-1) \tag{15}
$$
$$
(x,t) \in [0, 1] \times [0, 1]
$$

---

[1]https://github.com/dynamicslab/pysindy

# F HYPERPARAMETERS

Table 4: LLM hyperparameters

| Hyperparameter | Dataset | | | |
|---|---|---|---|---|
| | Burgers A | Burgers B | KdV | Wave |
| Iterations | 6 | 30 | 30 | 6 |
| Derivative order | [2, 3] | [2, 3] | [2, 3] | [2, 3] |
| Best candidates | 4 | 4 | 4 | 4 |

Due to the ongoing development of the EPDE framework, the results obtained with its newer versions may vary from those presented in this study. For these experiments, we use the hyperparameters presented in Table 5.

Table 5: EPDE hyperparameters

| Hyperparameter | Dataset | | | |
|---|---|---|---|---|
| | Burgers A | Burgers B | KdV | Wave |
| Epochs | 5 | 5 | 5 | 5 |
| Population size | 8 | 8 | 8 | 8 |
| Boundary | (20, 20) | (20, 50) | (40, 100) | (20, 20) |
| Derivative order | [2, 3] | [2, 3] | [2, 3] | [2, 3] |
| Term number | 5 | 5 | 5 | 5 |
| Function power | 3 | 3 | 3 | 3 |
| Sparsity interval | (1e-6, 1e-5) | (1e-6, 1e-5) | (1e-6, 1e-5) | (1e-6, 1e-5) |

# G   DETAILED EXPERIMENT PLOTS

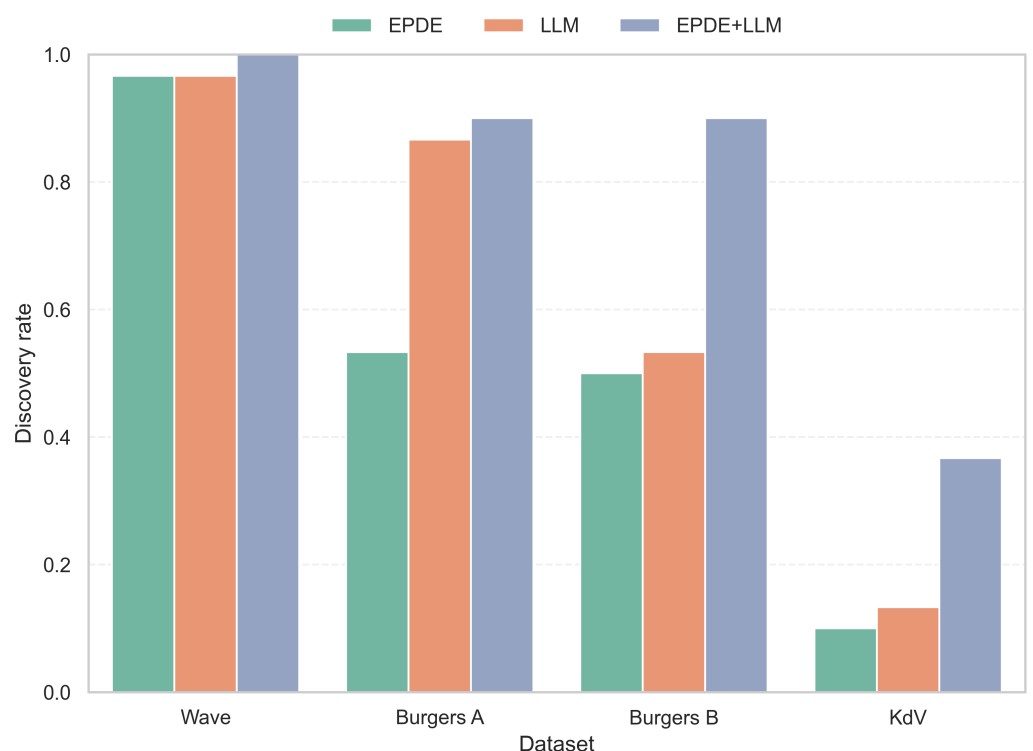

Figure 8: Comparison of discovery rates of the frameworks with clean data

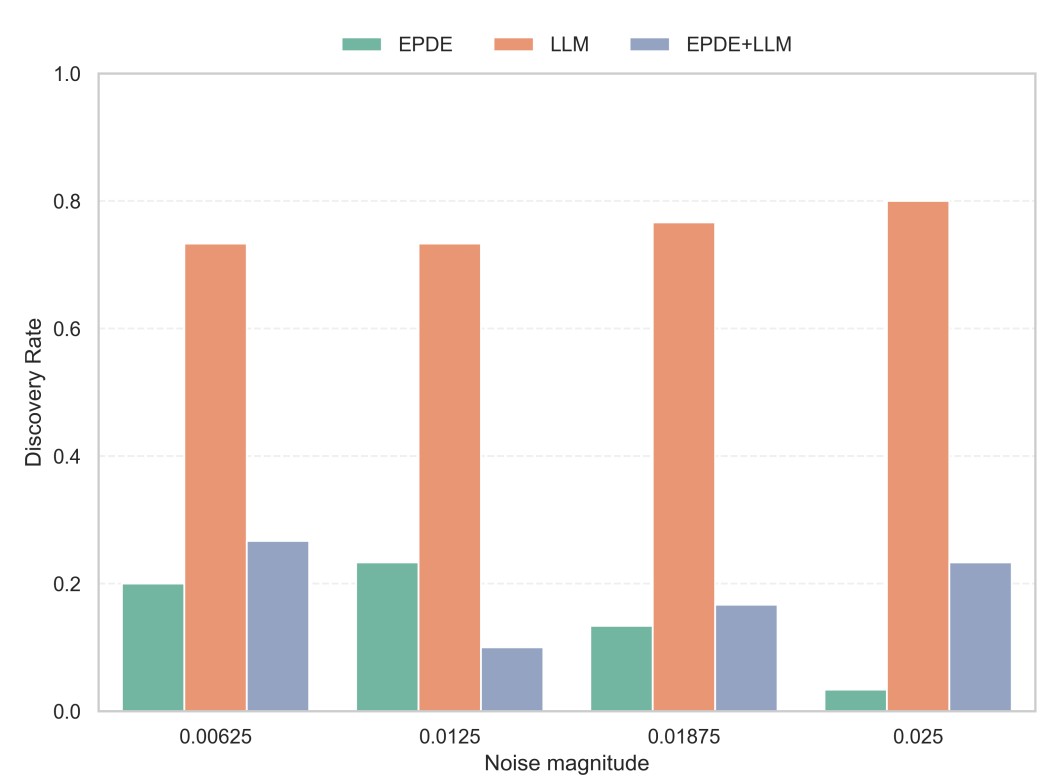

Figure 9: Comparison of discovery rates of the frameworks with noisy data – Burgers A

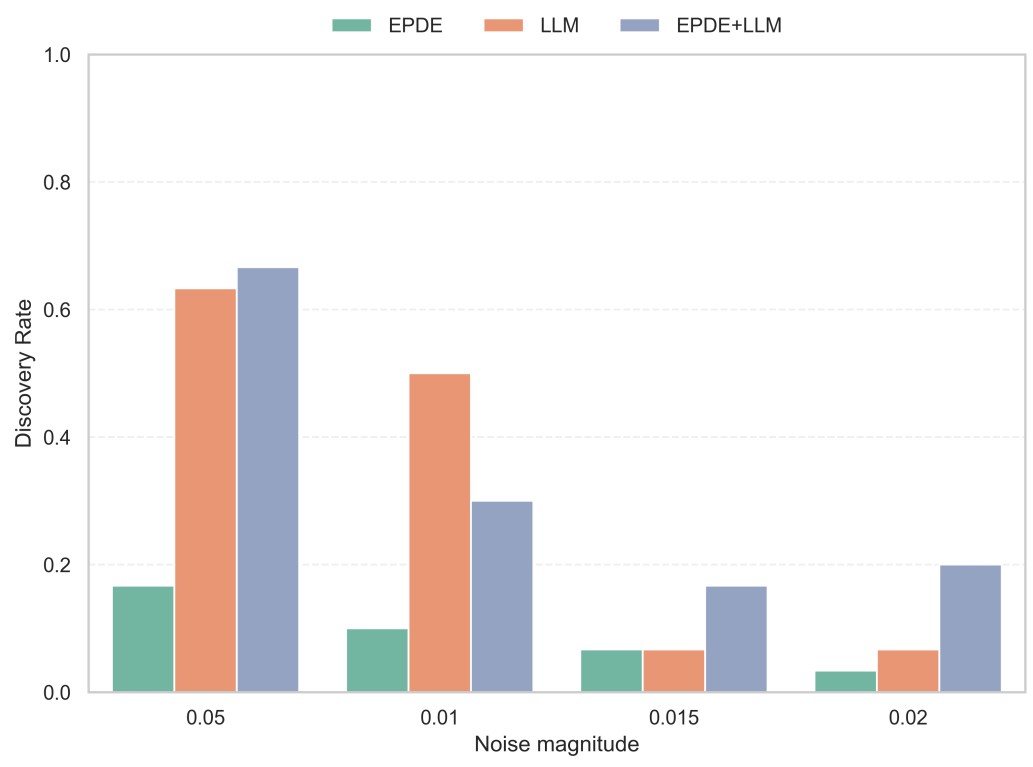

Figure 10: Comparison of discovery rates of the frameworks with noisy data – Burgers B

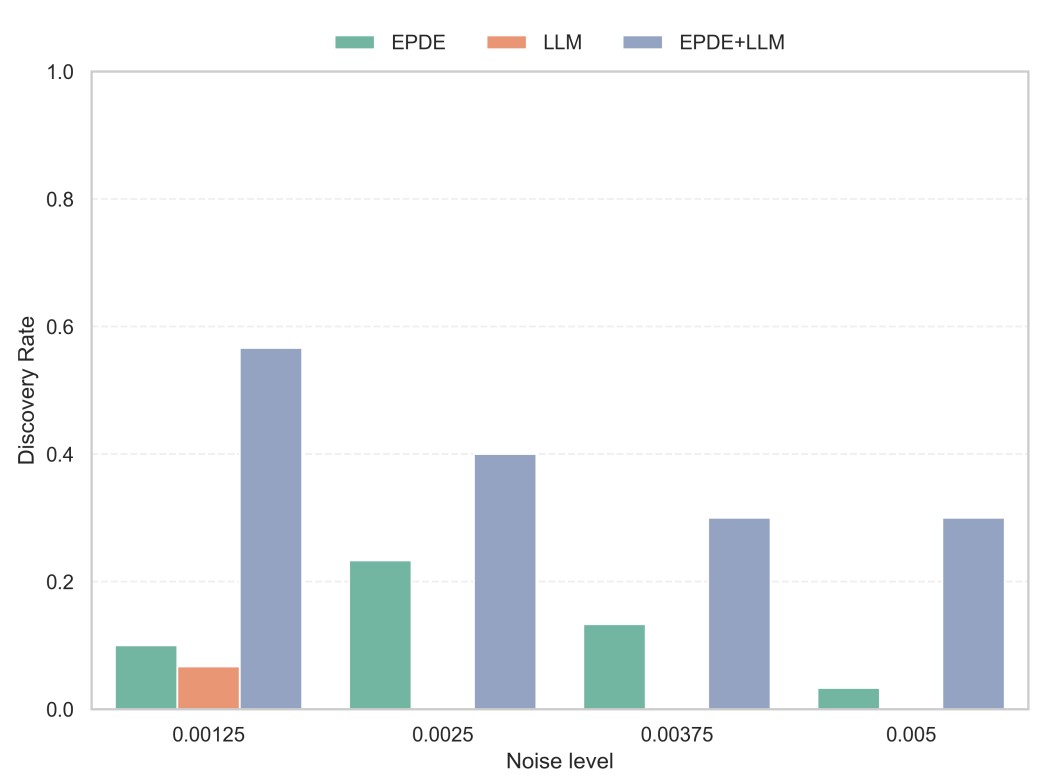

Figure 11: Comparison of discovery rates of the frameworks with noisy data – KdV equation

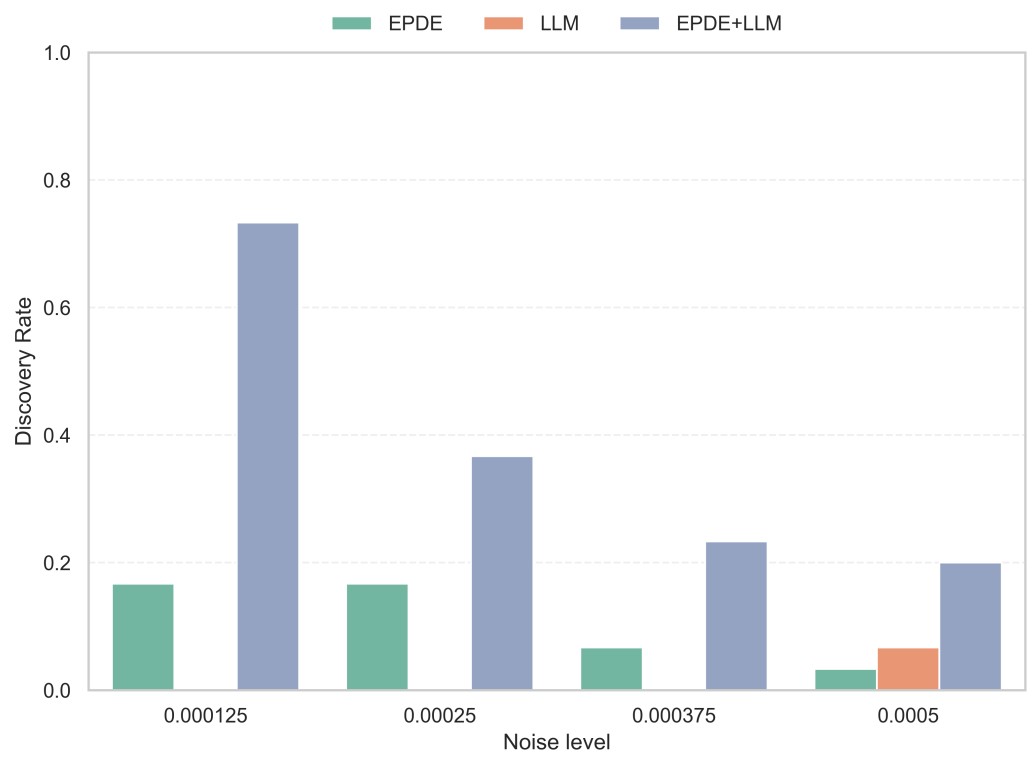

Figure 12: Comparison of discovery rates of the frameworks with noisy data – Wave equation

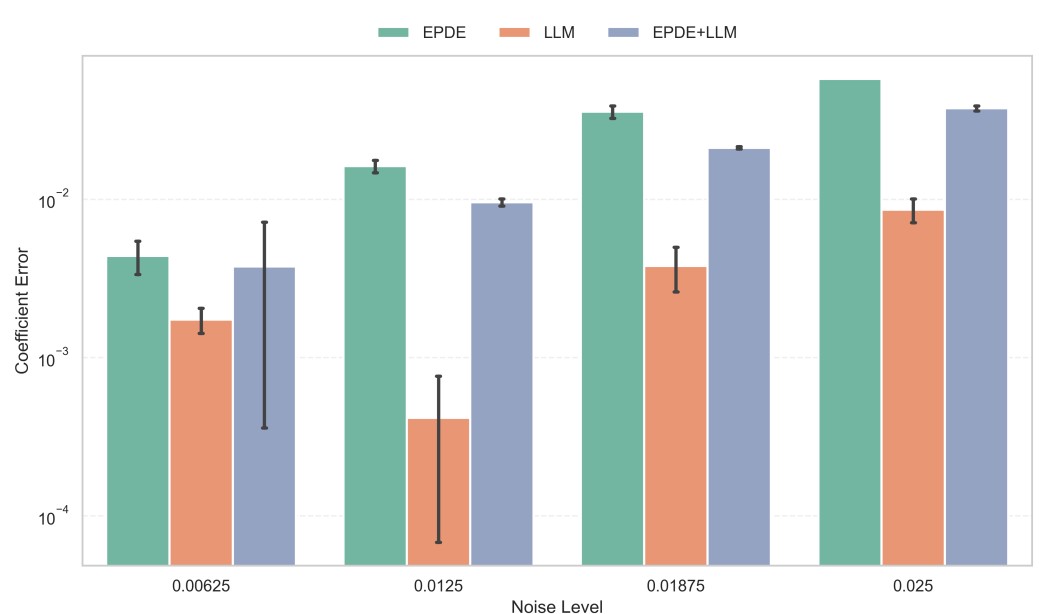

Figure 13: Comparison of coefficient errors of the frameworks with noisy data – Burgers A

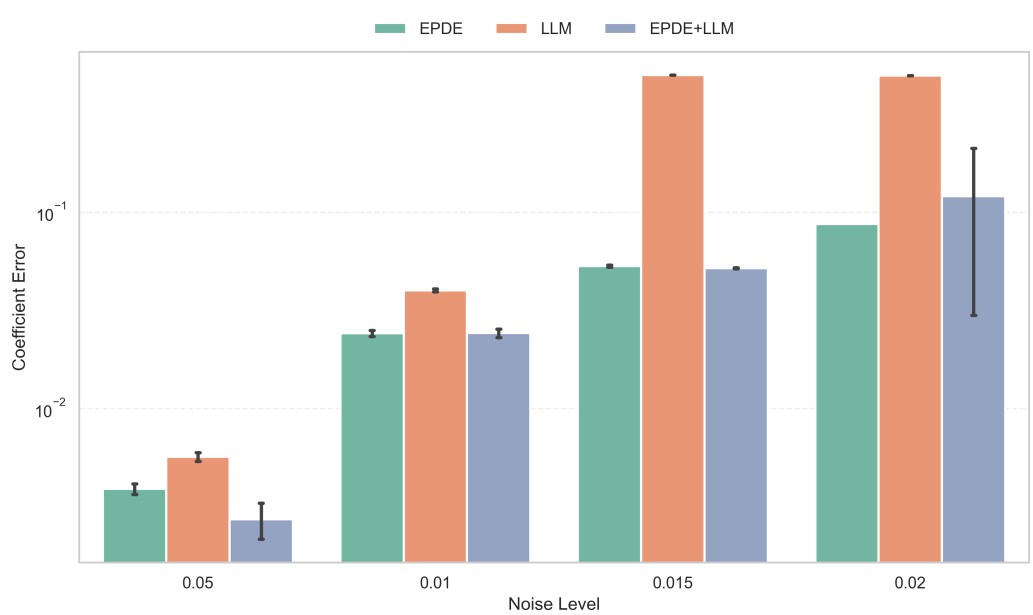

Figure 14: Comparison of coefficient errors of the frameworks with noisy data – Burgers B

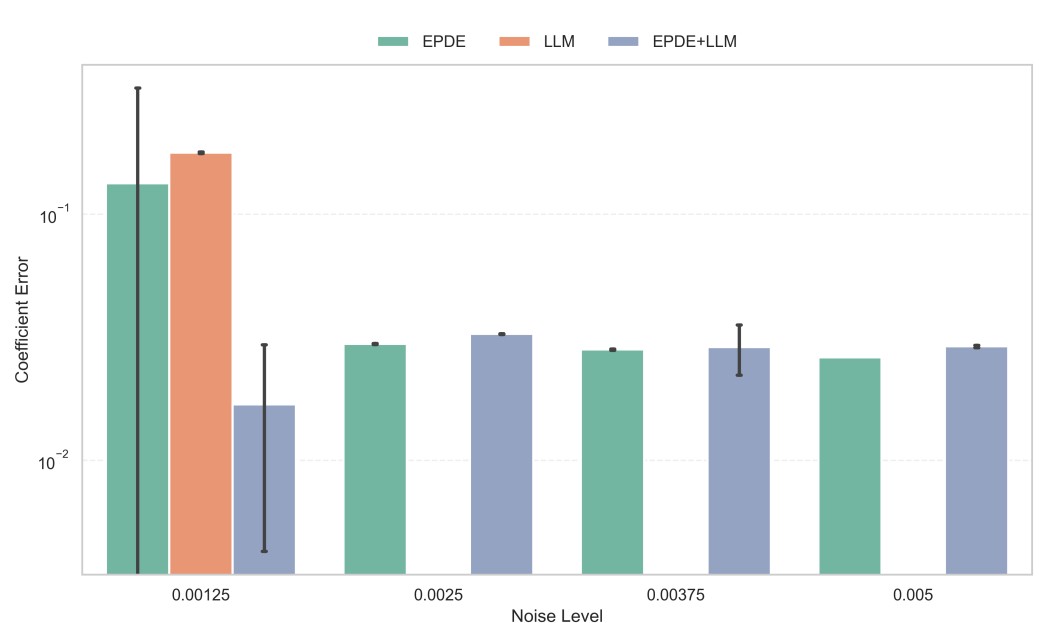

Figure 15: Comparison of coefficient errors of the frameworks with noisy data – KdV equation

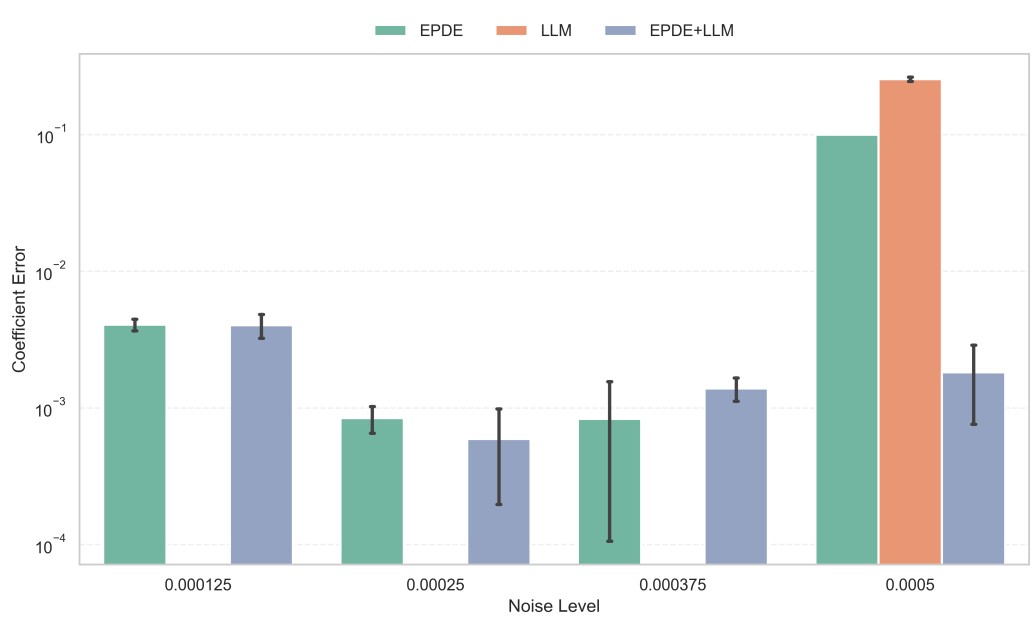

Figure 16: Comparison of coefficient errors of the frameworks with noisy data – Wave equation

