# OpenReview forum: "Does LLM dream of differential equation discovery?"
_ICLR.cc/2026/Conference — ICLR 2026 Conference Withdrawn Submission_

### Official Review · Reviewer_8Wqi · 2025-10-27

**Soundness:** 2
**Presentation:** 2
**Contribution:** 2
**Rating:** 2
**Confidence:** 3

**Summary:**

The paper performs PDE discovery by framing it as a code-generation task for LLMs. To do this they introduce a physics-preserving textual representation of the numerical field data that enables LLMs to infer symbolic PDE structures without requiring explicit solvers or symbolic differentiation. The work combines LLM-generated candidate equations with the EPDE (Evolutionary PDE) framework for structural refinement and coefficient fitting.

**Strengths:**

* Frames PDE discovery as a code generation task, enabling LLMs to infer symbolic equations from structured data.

* Combines LLM-generated equation candidates with EPDE’s evolutionary refinement for some improved discovery rates with low complexity.

**Weaknesses:**

* The authors perform prompt engineering with an LLM to suggest differentials that the EPDE uses to perform PDE discovery. However, the specific novel elements or enhancements introduced by the authors are not explicitly clarified.

* Table 2, 3: A visualization of the CE and DR tradeoff may help clarify the results / conclusions of this paper. Currently the tradeoffs in the two make it hard to determine which frameworks perform best overall.

> We formulated differential equation
 discovery as a code generation problem, and we develop an optimal format of data that enables
 LLM to extract the connection between differentials and data. The data must not be too compressed
 to retain physics, and on the other hand, must be compressed to fit the context.
* (L071 above): For an LLM input, it is not clear where the justifications for the paper's choice in compression balance is. Why is the list of values presented in sec D.3 - D.5 optimal?

* How many examples were considered for each PDE dataset? Details about the dataset seem to be missing or are unclear.

* Presentation / Readability: it would be helpful to the reader to add for example, more explanations in figure 2 / match the tittles to the text / add references to specific sections.

> To address this, we evaluated several strategies, including visual
 language models (VLMs), alternative data transformations, and tensor decomposition techniques.
 The most effective and viable solution was found to be a significant but careful dimensionality
 reduction. The original data was downsampled via interpolation to a coarse spatial resolution of
 approximately 20×20 to 30×30 grid points. This approach preserves the essential structural infor
 mation of the physical fields while drastically reducing token consumption, making the data tractable
 for LLM processing. A preliminary analysis of how VLMs handle such physical data was also con
 ducted.
* Sec 3.1 (above): it is unclear where are the justifications / experiments for certain claims: "The most effective and viable solution was found to be a significant but careful dimensionality reduction." Can the paper expand on the dimensionality reduction process and why this method for dimensionality reduction was chosen? What defines "optimal" in the tradeoff between token consumption and data compression? A reference to the VLM analysis would also aid readability.

**Questions:**

* How many iterations does the LLM need to generate reasonable results, are there diminishing returns, what are the variations in LLM suggestions?

* For a given PDE, is there a relationship between PDE characteristics (such as initial conditions, pde parameters coefficients) and the DR and CE?

---

### Official Review · Reviewer_U6vp · 2025-10-31

**Soundness:** 2
**Presentation:** 3
**Contribution:** 2
**Rating:** 4
**Confidence:** 5

**Summary:**

This paper discuss how LLM can aid in discovering PDEs from data by integrating LLMs as symbolic “oracles” within an EPDE framework, demonstrating that the hybrid EPDE+LLM pipeline outperforms either method alone in identifying correct equation structures and coefficients across clean and noisy datasets

**Strengths:**

1. Combines LLMs with EPDE, forming a hybrid framework for PDE discovery.
2. Demonstrates LLMs’ effectiveness as symbolic oracles via multi-framework comparison.

**Weaknesses:**

Here are my 3 major concerns:
1. The evaluation focuses only on classical PDEs like Burgers’, Wave, and KdV, which are widely present in physics textbooks and likely seen during LLM pre-training. The LLM’s success may partly stem from recalling known forms rather than discovering new ones from raw data. The paper would be stronger if it tested on synthetic or unseen PDEs beyond standard equations, clarifying generalization to novel physics.
2. While the hybrid generally outperforms both components, it fails on the noisy Burgers A dataset where the standalone LLM performs better. The paper attributes this to EPDE getting stuck in local minima, but this suggests the integration may sometimes discard useful LLM-generated hypotheses. This exposes a potential weakness: when the evolutionary stage is misled by noise, the hybrid pipeline can underperform despite good initial candidates. Further analysis of this case would strengthen the paper and guide more robust integration.
3. The novelty and contribution of the paper appear limited, and its main focus is not entirely clear. Some prior studies, such as [1], have already demonstrated the effectiveness of LLMs in PDE discovery, and several works have explored combining LLMs with genetic programming [1,2]. It is unclear what specific advantages this paper offers over these existing works. Moreover, the paper has no baseline comparisons with other methods. The discussion seems to focus primarily on the role of LLMs in PDE discovery — a topic that has already been well examined in earlier works — which makes this aspect of the contribution less compelling.
[1] Llm4ed: Large language models for automatic equation discovery
[2] Symbolic Regression with a Learned Concept Library

**Questions:**

Please refer to the weakness.

---

### Official Review · Reviewer_PFgm · 2025-11-01

**Soundness:** 2
**Presentation:** 2
**Contribution:** 2
**Rating:** 2
**Confidence:** 4

**Summary:**

This paper explores using LLMs for PDE discovery from numerical field data. Three approaches are tested: EPDE (evolutionary algorithm), LLM-only, and a hybrid EPDE+LLM pipeline. Experiments on Burgers, Wave, and KdV equations show the hybrid achieves highest discovery rates, though LLM-only exhibits significantly worse coefficient errors than EPDE-based methods.

**Strengths:**

* The formulation of PDE discovery as code generation and using LLMs for this task is interesting.
* The hybrid architecture showing how LLMs can effectively initialize evolutionary search provides useful empirical insights.
* The experimental protocol with several runs per configuration and measurement of both discovery rates and coefficient errors is good.

**Weaknesses:**

* **Evaluation Fairness:** The test equations (Burgers, Wave, KdV) are canonical PDEs that almost certainly appear in LLM training data, making it difficult to assess whether results demonstrate discovery capability versus pattern recognition/memorization by LLMs. Focusing on PDEs (rather than general function discovery) significantly limits the search space of realistic (and here well-known) PDE forms. With 30 LLM iterations, this raises the question of whether the approach is discovering equations or efficiently enumerating known forms.

* **Impaired baseline:** The LLM-only method's poor coefficient errors (10-100x worse than EPDE, Table 1) appear to stem from using downsampled data for both derivative computation and optimization, while EPDE uses full resolution, if I understand correctly. This coupling seems like an implementation choice rather than necessity; the LLM could generate structures from downsampled data (due to context length) but evaluate on full data.

* The downsampling to 20×20 grids seems quite aggressive for PDEs that might contain shocks, boundary layers, or multi-scale features, which limits the practical relevance.

* **Minor -- Presentation:** Writing quality could be improved throughout the paper. Some terminology (e.g., stability instead of complexity) are unclear. Reference formats are not correct, and the paper lacks appropriate referencing in many places (e.g., line 151, PIC).

**Questions:**

1. The test set PDEs are well-known equations. I'm curious if there could be any evidence that the LLM is performing discovery rather than retrieving memorized equations? For instance, some suggestion might include testing on novel or synthetic hybrid equations not in textbooks. Without this, it's unclear if the approach would work on truly unknown physics.

2. The PDE search space for these problems seems quite limited; perhaps <50 plausible forms considering common terms and combinations, and looking for well-studied, stable equations. With same number of LLM iterations plus feedback, what would a simple baseline achieve? For example: random sampling from a library of 50 template well-known PDEs from literature, or systematic enumeration with scoring. This would help quantify whether LLM broader knowledge provides advantage over simpler enumeration strategies.

3. For LLM-only, it seems that the derivatives and coefficient optimization both use downsampled (20×30) data, while EPDE uses full resolution. Why? Could the LLM generate structures from downsampled data (as input due to context length) but then compute derivatives and optimize coefficients on full-resolution data? This seems like it would eliminate the coefficient error gap. What prevents this more balanced comparison?

4. EPDE and EPDE+LLM show identical coefficient errors in table 1. Does this mean they use exactly the same optimization procedure on the same data?

5. It seems that the hybrid approach currently adds computational budget over LLM-only (LLM iterations + EPDE iterations). What if LLM-only ran for equivalent compute or number of iterations with full-resolution evaluation? Could additional LLM iterations be as effective as EPDE iterations?

6. Could you provide qualitative analysis of: (a) what initial populations the LLM generates versus random EPDE initialization, (b) the trajectory of equations discovered across iterations/generations, (c) examples where LLM proposes structures EPDE wouldn't find and vice versa? This would help understand the complementary strengths beyond aggregate metrics.

7. In figure 9, LLM-only outperforms EPDE+LLM. Is there a specific reason behind this? Does EPDE's evolution sometimes degrade LLM's good initial suggestions, or does it get stuck in local minima that LLM's broader sampling avoids?

---

### Official Review · Reviewer_sVao · 2025-11-02

**Soundness:** 1
**Presentation:** 1
**Contribution:** 1
**Rating:** 2
**Confidence:** 4

**Summary:**

This paper investigates the use of large language models (LLMs) for discovering partial differential equations (PDEs), combining symbolic generation from an LLM with refinement via the EPDE algorithm. The topic is timely, aiming to bridge symbolic reasoning with data-driven scientific discovery. However, the work lacks theoretical grounding, experimental rigor, and methodological clarity, which are necessary to meet ICLR standards.

**Strengths:**

+ Addresses an emerging and relevant question: how LLMs can assist PDE discovery.
+ Hybrid design combining symbolic and numeric reasoning is a meaningful direction in principle.
+ The study acknowledges limitations (single-equation discovery, grid-spaced data).
+ Provides an interesting observation that LLMs handle structured tabular input better than visual or raw array formats.

**Weaknesses:**

- The evaluation scope is extremely limited, testing only four simple PDEs (Wave, Burgers, KdV) with no standardized or real-world benchmarks such as PDEBench or APEBench.
- The assumption that LLMs inherently understand differential equations is speculative and unsupported by analysis or evidence.
- Comparisons to other modern PDE discovery methods are missing. There is no head-to-head evaluation against symbolic regression, neural operator, or PINN-based approaches.
- The methodology contains several arbitrary design choices. Downsampling to 20×20–30×30 grids likely distorts derivative estimation, yet its effect on discovery accuracy is never analyzed.
- The evaluation metrics and reporting are weak. No confidence intervals or statistical significance testing are provided despite small datasets and limited runs.
- The writing and organization are poor, with inconsistent notation, redundant figures, and unclear algorithmic descriptions.
- The experiments use a simplistic Gaussian noise model and ignore real-world challenges such as irregular sampling, boundary errors, and conservation-law constraints.
- The paper lacks ablations on key components such as the experience buffer, complexity metric, prompt structure, or knee detection process.
- Computational cost and runtime are not analyzed, leaving unclear whether the approach is practical at scale.

The conclusions overstate the results. Modest improvements on small synthetic datasets do not justify claims of robust scientific discovery.

The related work section is superficial and fails to situate this approach relative to recent PDE discovery frameworks like PDE-LEARN, DISCOVER, or neural operator-based models.

**Questions:**

Why were only four PDEs selected, and how would the method scale to coupled or higher-dimensional systems?
Why is the left-hand side fixed in all experiments? Have you tested discovery of full PDE structures?
How sensitive are results to grid resolution, noise level, and prompt formulation?
What prevents the LLM from generating arbitrary or spurious PDEs given only numerical data?
Why are there no comparisons with SINDy, PDE-LEARN, or neural operator baselines?
What is the runtime and computational cost (GPU hours, token count) for each method?
Can the method respect physical constraints or conservation properties in discovered equations?

---

### Note · Authors · 2025-12-03

**Comment:**

We are grateful to the reviewers for the valuable comments. We understand that our paper is yet not ready and require significant changes and therefore we decided to withdraw it to submit to another venue.

**Withdrawal Confirmation:**

I have read and agree with the venue's withdrawal policy on behalf of myself and my co-authors.